

# Nitrogen addition increases the contents of glomalin-related soil protein and soil organic carbon but retains aggregate stability in a *Pinus tabulaeformis* forest

Lipeng Sun[1],[*], Hang Jing[1],[*], Guoliang Wang[1],[2] and Guobin Liu[1],[2]

[1] Institute of Soil and Water Conservation, Northwest A&F University, Yangling, Shaanxi, China
[2] State Key Laboratory of Soil Erosion and Dryland Farming on the Loess Plateau, Institute of Soil and Water Conservation, Chinese Academy of Sciences and Ministry of Water Resources, Yangling, Shaanxi, China
[*] These authors contributed equally to this work.

Corresponding author
Guobin Liu, gbliu@ms.iswc.ac.cn

## ABSTRACT

**Background:** Glomalin-related soil protein (GRSP) and soil organic carbon (SOC) contribute to the formation and stability of soil aggregates, but the mechanism by which global atmospheric nitrogen (N) deposition changes soil aggregate stability by altering the distribution of GRSP and SOC in different aggregate fractions remains unknown.

**Methods:** We used a gradient N addition (0–9 g N m$^{-2}$ y$^{-1}$) in *Pinus tabulaeformis* forest for two years in northeast China and then examined the changes in SOC contents, total GRSP (T-GRSP), and easily extractable GRSP (EE-GRSP) contents in three soil aggregate fractions (macro-aggregate: >250 μm, micro-aggregate: 250–53 μm, and fine material: <53 μm) and their relationship with aggregate stability.

**Results:** (1) The soil was dominated by macro-aggregates. Short term N addition had no significant effect on mean weight diameter (MWD) and geometric mean diameter (GMD). (2) GRSP varied among aggregate fractions, and N addition had different effects on the distribution of GRSP in aggregate fractions. The EE-GRSP content in the macro-aggregates increased initially and then decreased with increasing N addition levels, having a peak value of 0.480 mg g$^{-1}$ at 6 g N m$^{-2}$ y$^{-1}$. The micro-aggregates had the lowest EE-GRSP content (0.148 mg g$^{-1}$) at 6 g N m$^{-2}$ y$^{-1}$. Furthermore, the T-GRSP content significantly increased in the aggregate fractions with the N addition levels. (3) The macro-aggregate had the highest SOC content, followed by the micro-aggregate and the fine material had the lowest SOC content. N addition significantly increased the SOC content in all the aggregate fractions. (4) GRSP and SOC contents were not significantly correlated with MWD.

**Conclusion:** Glomalin-related soil protein and SOC contents increased by N addition, but this increase did not enhance aggregate stability in short term, and the improvement of stability might depend on binding agents and incubation time.

## INTRODUCTION

Human activities have led to a rapid increase in global nitrogen (N) deposition (*Dirnböck et al., 2017*). The increasing N deposition has tremendous effects on soil quality. For example, N deposition can affect soil aggregate stability by changing plant growth rates (*Bai et al., 2010*), soil microbial activities (*Keeler, Hobbie & Kellogg, 2009*), and soil chemical properties in forest communities (*Yin et al., 2016*). However, the effects of N deposition on soil aggregate stability remains controversial (*Liu et al., 2015*; *Tripathi, Kushwaha & Singh, 2008*; *Zhong et al., 2017*), especially with respect to the mechanisms by which available N in soil influences aggregate stability when it changes glomalin-related soil protein (GRSP) and soil organic carbon (SOC) in aggregate fractions.

Glomalin-related soil protein can bind soil particles to form aggregates, and the stability of these aggregates may increase with increasing GRSP content (*Wright & Upadhyaya, 1998*). Several studies showed that soil available N has variable effects on GRSP content in bulk soil. For instance, some researchers found that N addition significantly increased GRSP content in bulk soil (*Zhang et al., 2015*; *Garcia et al., 2008*), but *Wuest et al. (2005)* found that N addition had no significant effects on GRSP content in bulk soil. These inconsistent results may be related to the soil initial N content (*Treseder, Turner & Mack, 2007*), as well as N addition level and duration (*Garcia et al., 2008*; *Treseder, 2004*). For example, N addition could increase the growth of arbuscular mycorrhizal fungi (AMF) in the soil with low initial N content (*Treseder & Allen, 2002*), thus it increases GRSP content in bulk soil because glomalin is mainly produced by AMF (*Treseder, Turner & Mack, 2007*). Meanwhile, N addition may inhibit the growth of AMF in soil with high initial N content and may thus decrease GRSP content in bulk soil (*Treseder, 2004*). However, a recent study showed that N addition reduced plant fine root length and AMF infection rate in soil with low N content (*Wang et al., 2017*), indicating that N addition can decrease AMF and GRSP content even in such soil (*Treseder, Turner & Mack, 2007*). Therefore, exploring the relationship between GRSP and available N content in soil with low N content is necessary.

Soil is mainly composed of aggregates of various sizes. As far as we know, the mechanism underlying the influence of N addition on GRSP content in aggregates of variable size has not been reported yet. In some studies on farmland ecosystems, manure amendment was found to promote soil nutrient accumulation and microbial activities, considerably increasing GRSP contents in all soil aggregate fractions except micro-aggregates (*Xie et al., 2015*). In these farmland ecosystems, tillage management methods disrupts soil structure and reduces microbial biomass and extent of enzyme activity, resulting in the significant decrease of GRSP content in all soil aggregate fractions except fine materials and small fractions of macro-aggregates (*Wright, Green & Cavigelli, 2007*). Similar to manure amendment and tillage management, N addition may also alter soil nutrient accumulation and microbial activity (*Treseder & Allen, 2002*; *Treseder, 2004*). Thus, N addition may change the GRSP content in soil aggregates. We speculated that the effects of N addition on GRSP content in aggregate fractions at forest system

might be similar to that of farmland ecosystems with manure amendment and tillage management, as both systems alter soil nutrient accumulation and microbial activity.

Glomalin-related soil protein can bind soil particles to form aggregates. Thus, changing of GRSP in aggregate fractions can influence aggregate stability directly (*Wright & Upadhyaya, 1998*). A large number of studies have shown that the correlation between GRSP and aggregate stability is significant and positive (*Wright, Green & Cavigelli, 2007*; *Zhang et al., 2014*). However, a recent study on the relationships between GRSP and aggregate stability in citrus rhizosphere found that easily extractable GRSP (EE-GRSP) in 0.25–0.5 mm aggregates was significantly positively correlated with aggregate stability, whereas the total GRSP (T-GRSP) in 2.0–4.0 mm aggregates and EE-GRSP in 0.5–1.0 mm aggregates was significantly negatively correlated with aggregate stability. However, the GRSP in other size aggregates (1.0–2.0 mm) was insignificantly correlated with aggregate stability (*Wu et al., 2013*). These results indicated that the relationships between GRSP and aggregate stability may vary with the aggregate sizes. Thus, we hypothesize that low-level N addition may increase the GRSP content in all soil aggregate fractions except micro-aggregates, and high-level N addition may decrease the GRSP content in all soil aggregate fractions except fine materials. The relationships between GRSP and aggregate stability will be different according to the aggregate sizes.

In most previous studies, N addition significantly increased SOC content in bulk soil (*Nave et al., 2009*), although some studies indicated that N addition considerably decreased or did not affect SOC content (*Waldrop et al., 2004*; *Zhong et al., 2017*). N addition has variable effects on SOC content in aggregate fractions and bulk soil (*Brown et al., 2014*; *Fonte et al., 2009*). However, most studies were focused on the effects of N addition on SOC content in bulk soil, and changes in SOC content in aggregate fractions are rarely investigated (*Pan et al., 2007*). In agriculture ecosystems, N fertilization has a minimal effect on SOC content in aggregate fractions under a conventional maize cropping system (*Brown et al., 2014*). In maize cropping systems utilizing organic residues, N addition significantly increases SOC content in fine materials and has no significant effect on the SOC content in other size aggregates (*Fonte et al., 2009*). These inconsistent results may be explained by the following reasons: first, the physical protection on SOC were different in aggregate fractions, thus resulting in different accumulation and decomposition rates of SOC in aggregate fractions (*Blanco-Canqui & Lal, 2004*). Second, tillage management affects soil carbon (C) input. In a conventional maize cropping system, N addition has no significant effect on SOC content in aggregate fractions, but it has considerable effect on SOC content in fine materials in a maize cropping system utilizing organic residues. The increased SOC content in fine materials may be due to the input of organic residues (*Fonte et al., 2009*). At present, studies on the effect of soil available N on SOC in aggregates are mainly focused on farmland ecosystems, and no study on forestland system has been reported yet (*Chen et al., 2017*). We speculated that the effects of N addition on SOC content in aggregate fractions at forest system might be similar to that of maize cropping systems utilizing organic residues, as both systems return a large amount of plant biomass to soil. This speculation requires experimental validation. Recent studies found that the relationship between

SOC and aggregate stability is uncertain. For example, SOC content has a significantly negative relationship with aggregate stability in red soil found in subtropical China (*Li et al., 2005*). Moreover, the relationships between SOC and aggregate stability were insignificant in different land use tropical Ultisol (*Leelamanie & Mapa, 2015*). SOC content in aggregate fractions had different relationships with aggregate stability in rice–wheat rotation farmlands (*Das et al., 2014*). For example, SOC content in macro-aggregates is significantly positively correlated with aggregate stability, but SOC content in micro- and fine materials significantly negatively correlated with aggregate stability. However, research on the relationships between SOC content in aggregate fractions and aggregate stability under N addition condition remains lacking. In the present study, we proposed that N addition significantly increases SOC content in fine materials and minimal affect SOC content in other size aggregate fractions. The relationships between SOC content and aggregate stability are varied among aggregate fractions.

A total of two hypotheses were tested by determining GRSP and SOC contents in aggregate fractions and aggregate stability in a *Pinus tabulaeformis* forestland. This study is expected to clarify on the relationship between GRSP in different soil aggregate sizes and aggregate stability for the sake of N deposition in this region.

## MATERIALS AND METHODS

### Study area

The experiment was conducted in the Songyugou watershed in the Loess Plateau region, Shaanxi Province, China (35 390N, 110 060E), the elevation and slope of the study area were 1,000–1,200 m and 20°–25°. The region has a classic mainland monsoon-type climate with a mean annual precipitation of 584.4 mm, mean annual temperature of 9.7 °C, and frost-free period of 180 days. The soil is gray forest soil (Gray Luvisols, FAO soil classification) with a surface soil (0–20 cm) bulk density of 1.1 g/cm$^3$. The soil total phosphorus (P) content is 1.42 ± 0.38 g/kg, and the soil pH is 8.6. *P. tabulaeformis* was planted in 1960. The current stand density is 1,400–1,800/ha. The average canopy density is 0.7. The average diameter at breast height is 10.0 cm. The average height of the tree is 11.2 m. The forest stand volume is 75.5 m$^3$/ha. The leaf area index is 6.34. The biomass of tree layer, shrub layer, and herbaceous layer are 112.96, 3.56, and 8.28 t/hm$^2$, respectively, the plant diversity index of the community is 0.51 (calculated using the Simpson method). Shrubs are mainly *Elaeagnus pungens* Thunb, *Rosa xanthina* Lindl, *Spiraea Salicifolia* L, *Lonicera japonica* Thunb, and *Viburnum dilatatum* Thunb. The shrub coverage is 30%. Herbs are mainly *Carex lanceolata* Boott, and the herb coverage is 30–50%.

### Experimental design

In China, the N deposition rate in 2010 was ranged from 0.1 to 7.43 g N m$^{-2}$ y$^{-1}$, with an average of 2.11 g N m$^{-2}$ y$^{-1}$ (*Liu et al., 2013*). The ambient N deposition rate in the region was approximately 1.2 g N m$^{-2}$ y$^{-1}$, and N deposition is increasing at an annual rate of 0.041 g N m$^{-2}$ y$^{-1}$ (*Lü & Tian, 2007*). Moreover, most studies added 0–12 g N m$^{-2}$ y$^{-1}$, which is sufficient to change soil N content from limitation to

saturation for plant growth in most ecosystems (*Wang et al., 2018*; *Jing et al., 2017a*). Thus, we designed four levels of N addition treatments (0, 3, 6, and 9 g N m$^{-2}$ y$^{-1}$) in the form of urea (Fumin Agriculture Product Company, Xi'an, China). In each treatment, six 10 × 10 m plots were randomly established for N addition. A total of 24 plots were subjected to the same climate, stand age, and terrain conditions. In 2014 and 2015, urea was dissolved in 10 L distilled water and evenly sprayed in each plot before the rain in early April, June, August, and October every year. Soil samples were collected after N addition in September 2015, and N was added for two years.

## Soil sampling

In September 2015, soil was collected from 0 to 20 cm soil layer (remove litter layer), because the short-term N addition directly affected the physical and chemical properties of the surface soil. Three undisturbed soil samples were randomly collected in each plot. The undisturbed soil samples were carefully transported back to the laboratory. The samples were gently broken apart and all the soil samples were air-dried after passed through 8 mm sieve. The three soil samples of each plot were mixed up after air-drying. Finally, all soil samples were sealed in plastic bags without squeezing until physical and chemical analysis.

## Aggregate distribution

Soil aggregates were classified by wet sieving method (*Six et al., 1998*). Air-dried samples (100 g) placed in a 250 μm sieve and submerged in deionized water for 5 min, and the surface of the water submerged the soil samples. Aggregates were separated by moving the sieve up and down for 3 cm with 50 repetitions/min for 2 min. Aggregates retained in the sieve (>250 μm) were macro-aggregates. The soil and water passed through the 250 μm sieve and then were sieved again as above using a 53 μm sieve for 3 min. Aggregates retained in the sieve (>53 μm) were micro-aggregates, and those that passed through the sieve were fine materials. All separated soil and water were collected, oven-dried at 45 °C, and weighed and stored for chemical analysis.

## GRSP analyses

Glomalin-related soil protein content was determined according to the method of *Wright & Upadhyaya (1998)*. T-GRSP was extracted from soil sample (1 g) with 8 mL of 50 mmol/L citrate solution at pH 7.0 and then autoclaved at 121 °C for 60 min. The supernatant was removed after centrifugation at 10,000×g for 10 min. This procedure was repeated five times on the same sample until the solution was straw-colored. The supernatant was pooled together and stored at 4 °C. EE-GRSP was extracted from soil sample (1 g) with 8 mL of 20 mmol/L citrate solution at pH 7.0 and autoclaved at 121 °C for 30 min. The supernatant was removed after centrifugation at 10,000×g for 10 min and stored at 4 °C. The T-GRSP or EE-GRSP in the supernatant was determined by Bradford analysis spectrophotometrically at 590 nm light (*Wright & Upadhyaya, 1996*; *Rillig, 2004*; *Wright & Upadhyaya, 1996*).

## SOC analyses

Soil organic carbon content was measured by dichromate oxidation procedure (*Nelson & Sommers, 1982*).

## Statistical analyses

Aggregate stability was evaluated by mean weight diameter (MWD) millimeter and geometric mean diameter (GMD) millimetr. MWD is a comprehensive index for the evaluation of aggregate stability, and GMD is an index of the main grain size distribution of aggregates. The larger value of MWD and GMD indicate higher aggregate stability (*Bedini et al., 2009*). MWD and GMD were calculated using the following equation:

$$\text{MWD} = \sum_{i=1}^{n} x_i w_i \quad \text{and} \quad \text{GMD} = \text{EXP}\left[\frac{\sum_{i=1}^{n} w_i \ln x_i}{\sum_{i=1}^{n} x_i}\right]$$

$x_i$ is the mean diameter of each class aggregate (mm), $w_i$ is the percentage of each class aggregate (%), and $n$ is the number of aggregate fraction classes.

Differences between any two N addition treatments in aggregate percentage, MWD, GMD, GRSP content, and SOC content were tested with one-way ANOVA on SPSS 20.0 statistical software package (SPSS Inc., Chicago, IL, USA). Linear regression analyses were used to test the relationships among GRSP, SOC, and MWD.

# RESULTS

## Effects of N addition on soil aggregate distribution and stability

The percentage of aggregate varied with size classes, and macro-aggregates were more dominant than the other size aggregate fractions in all samples (Table 1). Across all N addition treatments, the values of MWD were in the range of 1.38–1.70 mm, and GMD was in the range of 0.86–1.05 mm. N addition had no significant effects on aggregate distribution, MWD, and GMD. However, the percentage of macro-aggregate, MWD, and GMD increased first and then decreased with increasing N addition levels, whereas the percentage of micro-aggregate and fine material decreased first and then increased with increasing N addition levels.

## Effects of N addition on GRSP content

Figure 1 shows that the contents of EE-GRSP and T-GRSP were in the range of 0.099–0.551 and 0.315–4.058 mg g$^{-1}$, respectively. In the control treatment, GRSP content varied among bulk soil and aggregate fractions, and GRSP content increased in the order of macro-aggregate<bulk soil<micro-aggregate<fine material. Across all N addition treatments, EE-GRSP content in the macro-aggregates and bulk soil increased first and then decreased with the increasing N addition levels and had a top value at 6 g N m$^{-2}$ y$^{-1}$. Compared with that of control treatment, N addition changed the EE-GRSP content of the macro-, micro-, fine materials, and bulk soil by −24.8% to 222.1%, −30.8% to 22.0%, 22.8–40%, and −8.4% to 134.4%. Across all N addition treatments, T-GRSP contents in all size aggregates and bulk soil increased with the increasing N addition levels and had a top value at 9 g N m$^{-2}$ y$^{-1}$. Compared with that of control treatment, N addition

**Table 1 The soil aggregate characteristic of different N addition treatments.**

| Treatment (g N m$^{-2}$ y$^{-1}$) | Aggregate proportion in size class (%) | | | MWD (mm) | GMD (mm) |
|---|---|---|---|---|---|
| | >0.25 mm | 0.053—0.25 mm | <0.053 mm | | |
| CK(0) | 65.36 ± 7.63[a] | 28.49 ± 6.40[a] | 6.16 ± 1.98[a] | 1.45 ± 0.073[a] | 0.90 ± 0.039[a] |
| N3(3) | 68.78 ± 7.99[a] | 25.95 ± 7.41[a] | 5.28 ± 0.72[a] | 1.50 ± 0.158[a] | 0.94 ± 0.086[a] |
| N6(6) | 73.65 ± 5.05[a] | 21.88 ± 4.57[a] | 4.48 ± 0.90[a] | 1.60 ± 0.101[a] | 0.99 ± 0.063[a] |
| N9(9) | 72.76 ± 7.82[a] | 22.11 ± 6.45[a] | 5.14 ± 1.48[a] | 1.58 ± 0.156[a] | 0.98 ± 0.099[a] |

Notes:
Values followed by a different letter indicate significant difference among the treatments.
MWD, mean weight diameter; GMD, geometric mean diameter.
$P < 0.05$, $n = 6$.

changed the T-GRSP content in macro-, micro-, fine materials, and bulk soil by 412.7–872.4%, 203.1–266.9%, −0.3% to 14.3%, and 259.3–468.0%.

### Effects of N addition on SOC content

Soil organic carbon content ranged from 6.478 to 21.627 mg g$^{-1}$. Across all aggregate fractions, macro-aggregate had the highest content of SOC and fine material had the lowest content of SOC (Fig. 2). Compared with that of control treatment, N addition significantly increased SOC content in all size aggregate fractions and bulk soil, and the change ranges of SOC content in the macro-, micro-, fine materials, and bulk soil were 71.7–79.5%, 75.5–91.5%, 49.6–53.5%, and 47.6–74.4%.

### Relationships among GRSP, SOC and MWD

Table 2 shows that the contents of SOC, EE-GRSP and T-GRSP in aggregate fractions and bulk soil have no significant relationship with MWD (except for the relationship between SOC content in micro-aggregates and MWD).

## DISCUSSION

In this study, the soil of *P. tabulaeformis* forest was dominated by macro-aggregates, and the aggregate stability indexes of MWD and GMD were in the range of 1.38–1.70 and 0.86–1.05 mm, respectively. N addition for two years had no significant effects on the soil aggregate stability indexes, which was consistent with those of the previous studies (*Wang et al., 2013*; *Zhong et al., 2017*). For example, Zhong found that N addition for one year insignificantly changed the soil aggregate distribution in subtropical forest. Although N addition had no significant effect on aggregate distribution and stability, the percentage of macro-aggregate and values of MWD and GMD were higher in N addition treatments compared with that in control treatment. Meanwhile, N addition changed the contents of GRSP and SOC in aggregate fractions, which indicated that N addition could have a trend to change the soil aggregate distribution and could increase the aggregate stability with N addition duration.

### Effects of N addition on GRSP content

The contents of EE-GRSP and T-GRSP were in the range of 0.099–0.551 and 0.315–4.058 mg g$^{-1}$, respectively. This finding is consistent with that of the studied on the GRSP

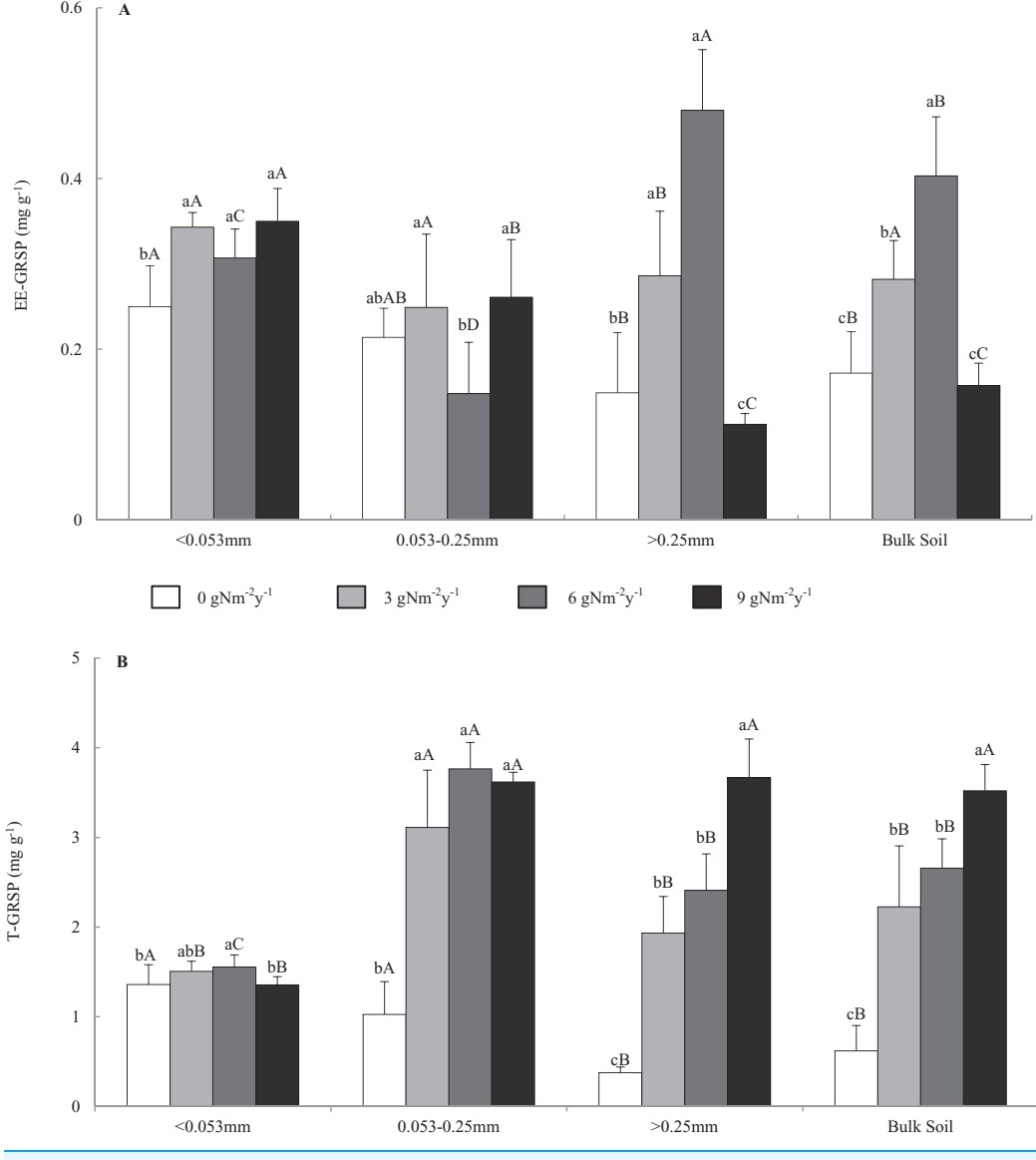

**Figure 1 Distributions of easily extractable and total glomalin-related soil protein (A) EE-GRSP and (B) T-GRSP content in soil aggregates and bulk soil of different N addition treatments.** Values followed by a different lowercase letters indicate significant difference among the N addition treatments, values followed by a different uppercase letters indicate significant difference among the aggregate size. $P < 0.05$, $n = 6$.               

contents of different vegetation communities (0.6–5.8 mg g$^{-1}$) (*Singh, Singh & Tripathi, 2013*). In this study, GRSP content varied among aggregate fractions and bulk soil, and N addition had different effects on GRSP distribution in aggregate fractions and bulk soil. The research in farmland system found that tillage management and manure amendment had different effects on GRSP distribution in aggregate fractions (*Wright, Green & Cavigelli, 2007*; *Xie et al., 2015*). Long-term manure amendments redistributed the GRSP in the macro-aggregate fractions. These results indicated that GRSP distribution in aggregate fractions is susceptible to environmental changes. EE-GRSP was produced by

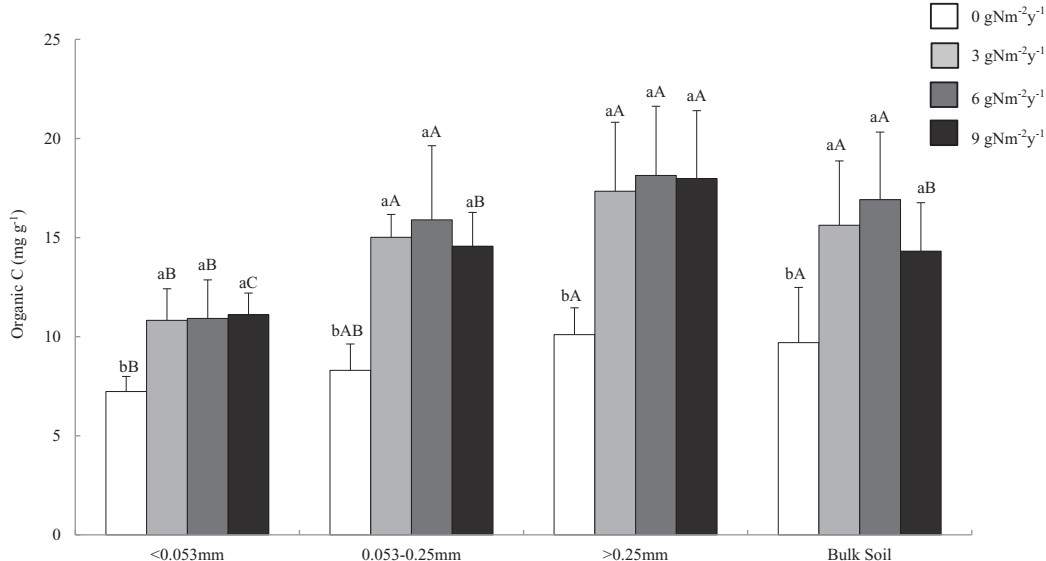

**Figure 2 Distributions of organic C content in soil aggregates and bulk soil of different N addition treatments.** Values followed by a different lowercase letters indicate significant difference among the N addition treatments, values followed by a different uppercase letters indicate significant difference among the aggregate size. $P < 0.05$, $n = 6$.

**Table 2 Bivariate correlations ($r$ values) between the mean weight diameter (MWD) and soil organic carbon (SOC), easily extractable and total glomalin-related soil protein (EE-GRSP and T-GRSP).**

|  | SOC | EE-GRSP | T-GRSP |
|---|---|---|---|
| Bulk soil | 0.381 | 0.176 | 0.250 |
| >0.25 mm | 0.254 | 0.156 | 0.266 |
| 0.25—0.053 mm | 0.420* | −0.050 | 0.318 |
| <0.053 mm | 0.376 | 0.098 | −0.006 |

Note:
 Asterisk indicates a significant correlation, $P < 0.05$, $n = 24$.

AMF most recently, and EE-GRSP was proved to be more sensitive to environmental changes than T-GRSP (*Singh, Singh & Tripathi, 2013*). In our study, the content of EE-GRSP in bulk soil increased first and then decreased with the increasing N addition levels and reached the peak (0.403 mg g$^{-1}$) at 6 g N m$^{-2}$ y$^{-1}$. The variation in EE-GRSP content was mainly attributed to the changes in plant growth and soil microbial activities (*Xie et al., 2015*), which were induced by N addition. In our study, plant growth and soil microbial activities might be increased at 3 g N m$^{-2}$ y$^{-1}$ and inhibited at 9 g N m$^{-2}$ y$^{-1}$. Thus, N addition of 6 g N m$^{-2}$ y$^{-1}$ might be the threshold value between increased and inhibited GRSP production. *Xie et al. (2015)* found that the highest EE-GRSP contents were not obtained at the highest amount of organic manure. Our results indicated that low-level N addition could increase the EE-GRSP contents, but high-level N addition inhibited the EE-GRSP contents. The highest EE-GRSP contents were observed at 6 g N m$^{-2}$ y$^{-1}$, which indicated a proper N addition level can maximize the EE-GRSP content in this region. These results supported our first hypothesis.

Furthermore, N addition had different effects on EE-GRSP content in aggregate fractions. The EE-GRSP content in macro-aggregates increased initially and then decreased with the increasing N addition levels and had a peak value at 6 g N m$^{-2}$ y$^{-1}$. Meanwhile, the EE-GRSP content in micro-aggregates had a low value at 6 g N m$^{-2}$ y$^{-1}$. These results were inconsistent with our first hypothesis. Previous studies showed that organic manure significantly increased the EE-GRSP content in all soil aggregate fractions except micro-aggregates in farmland ecosystems (*Xie et al., 2015*), and increasing soil depth significantly increased the EE-GRSP content in 0.5–1.0 and 1–2 mm aggregates without other size aggregate fractions in citrus rhizosphere (*Wu et al., 2013*). These different results indicated that the distribution of EE-GRSP in aggregate fractions varied with environmental conditions, which resulted in the changes of EE-GRSP in bulk soil. In addition, the percentage of macro-aggregates was higher than those of other size aggregate fractions. N addition has the same effects at EE-GRSP content in macro-aggregates with that in bulk soil, which indicated that the EE-GRSP in macro-aggregates was the main factor that determined the EE-GRSP in bulk soil. T-GRSP contained easily and hardly extractable parts. Compared with EE-GRSP, T-GRSP can reflect the accumulation of glomalin in the soil (*Singh, Singh & Tripathi, 2013*). In our study, N addition significantly increased the T-GRSP content in bulk soil and had a peak value at 9 g N m$^{-2}$ y$^{-1}$, which was different from the changes of EE-GRSP content in bulk soil. A large number of studies found that environmental changes had different effects on T-GRSP and EE-GRSP contents in bulk soil as well (*Wright, Green & Cavigelli, 2007*; *Xie et al., 2015*; *Antibus et al., 2006*; *Zhang et al., 2015*). An explanation for this result in our study is that high-level N addition can quickly relieve N limitation in the soil and promote the microbial activities at the beginning of the experiment, which increase GRSP production in the short term. Given the increasing time, high-level N addition may cause N saturation in the soil and inhibit microbial activities that decrease the GRSP production (*Yu et al., 2013*). However, the T-GRSP content under high N addition level was higher than that under low N addition level for a period of time due to the cumulative effect. The T-GRSP content in aggregate fractions increased significantly with the N addition, this result supports our explanation. While the continued responses of T-GRSP depend on the time and rate of N addition. Another explanation is that the conventional Bradford method we used in this study is often hardly to separate T-GRSP from total protein (*Roberts & Jones, 2008*). So continuing the study for three to five more years with infrared spectroscopy for analysis of the glomalin (*Jiří et al., 2017*) would be very essential. Besides, the micro-aggregates had the highest contents of T-GRSP in our study, which was consistent with previous research in this region (*Jing et al., 2017b*). However the distribution of T-GRSP in aggregates was variable based on different tillage management treatments in farmland (*Dai et al., 2015*). These results suggest that the distribution of GRSP in aggregates was affected by soil types, vegetation, climate, and other factors.

**Effects of N addition on SOC content**

In this study, SOC content ranged from 6.478 to 21.627 mg g$^{-1}$. Across all aggregate fractions, macro-aggregate had the highest SOC content, followed by the micro-aggregate,

and those in fine materials is the lowest. These results were consistent with that of the previous study (*Zhang et al., 2014*). However, some studies showed that the highest SOC content was in fine material, and the lowest was in macro-aggregate (*Gao et al., 2013*; *Tang et al., 2016*). Different results might be caused by soil type, plant community, and experimental conditions (*Wei et al., 2011*). N addition for two years significantly increased SOC content in bulk soil, which indicated that N addition could quickly increase plant growth and decomposition in N limitation soil and increase soil C storage (*Nave et al., 2009*). Interestingly, N addition significantly increased SOC content in all size aggregate fractions. Compared with other size aggregate fractions, macro-aggregate had the highest percentage and SOC content, with the greatest contribution to the soil C storage. Meanwhile, the SOC in fine material had the smallest contribution to the soil C storage. These results were different from our hypothesis. *Zhong et al. (2017)* found that N addition (0–300 kg N ha$^{-1}$ y$^{-1}$) for one year had no significant influence on SOC in any size aggregate fractions in a subtropical forest. These different results might be related to soil type and initial N content. For example, macro- and micro-aggregate fractions had the highest SOC content among the size aggregate fractions in loamy soils, while fine material fraction had the highest SOC content than those of other size aggregate fractions in sandy soil (*Gao et al., 2013*). Our results indicated that the SOC content varied among aggregate fractions, and N addition had different effects on SOC in aggregate fractions. The effects were not only affected by N addition level but also by soil type and initial N content.

## Relationships among GRSP, SOC and aggregate stability

GRSP and SOC were insignificantly correlated with MWD. This result fails to support our hypothesis. A large number of studies found that GRSP and SOC were significantly correlated with MWD because both can bind soil particle for aggregate formation (*Rillig, 2004*; *Wright, Green & Cavigelli, 2007*; *Baldock, Kay & Schnitzer, 1987*). However, recent studies found that GRSP and/or SOC may have no significant relationship with MWD (*Wu et al., 2013*; *Li et al., 2005*; *Leelamanie & Mapa, 2015*) because aggregate stability was not only affected by GRSP and SOC contents but also by soil texture, human disturbance, and other factors (*Wei et al., 2011*; *Tang et al., 2016*). In our study, the contents of GRSP and SOC significantly changed, but MWD insignificantly changed with N addition levels which caused insignificant relationships among MWD, GRSP and SOC. N addition significantly increased the contents of GRSP and SOC and insignificantly increased MWD, indicating that increasing GRSP and SOC contents does not necessarily increase aggregate stability rapidly. Aggregate formation requires not only binding agents but also physicochemical effects, but the underlying mechanism still needs further study.

## CONCLUSION

Nitrogen addition has different effects on GRSP content in aggregate fractions at *P. tabulaeformis* forestland. The EE-GRSP content in macro-aggregates increased initially and then decreased with the increasing N addition levels and had a peak value at 6 g N m$^{-2}$ y$^{-1}$, indicating a proper N addition level might maximize the GRSP production

in this region. The T-GRSP content in aggregate fractions increased significantly with the N addition and had a peak value at 9 g N m$^{-2}$ y$^{-1}$, because of the cumulative effect. Compare with control treatment, SOC content in all size aggregate fractions significantly increased in N addition treatments, which indicate that N addition could quickly increase plant growth in N limitation soil and increase soil C storage. However, N addition for two years has no significant effects on MWD and GMD, and GRSP and SOC are insignificantly correlated with MWD. This study identifies that the underlying mechanism of aggregate formation requires binding agents and reaction time both. Increasing GRSP and SOC contents does not necessarily increase aggregate stability rapidly.

### Funding

This work was supported by the National Natural Science Foundation of China (No. 41671513), and the National Key Research and Development Program of China (2017YFC0504601). The funders had no role in study design, data collection and analysis, decision to publish, or preparation of the manuscript.

### Grant Disclosures

The following grant information was disclosed by the authors:
National Natural Science Foundation of China: 41671513.
National Key Research and Development Program of China: 2017YFC0504601.

### Competing Interests

The authors declare that they have no competing interests.

### Author Contributions

- Lipeng Sun conceived and designed the experiments, performed the experiments, analyzed the data, contributed reagents/materials/analysis tools, prepared figures and/or tables, approved the final draft.
- Hang Jing performed the experiments, approved the final draft.
- Guoliang Wang conceived and designed the experiments, authored or reviewed drafts of the paper, approved the final draft.
- Guobin Liu conceived and designed the experiments, authored or reviewed drafts of the paper, approved the final draft.

### Data Availability

The raw data are provided in a Supplemental File.

### Supplemental Information

Supplemental information for this article can be found online at http://dx.doi.org/10.7717/peerj.5039#supplemental-information.

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
