# Peer review of "Nitrogen addition increases the contents of glomalin-related soil protein and soil organic carbon but retains aggregate stability in a Pinus tabulaeformis forest"

_PeerJ, doi:10.7717/peerj.5039_

## Round 0.1 · original submission · Major Revisions

Dear Dr. Liu,

We have received peer reviews of your manuscript entitled "Nitrogen addition increases the contents of glomalin-related soil protein and soil organic carbon but retains aggregate stability in a Pinus tabulaeformis forest", which was submitted to PeerJ. Based on these reviews, your manuscript may be accepted for publication after you have addressed the comments and corrections suggested by the reviewer(s).

When preparing your revised manuscript, you are asked to carefully consider the reviewer comments that are attached, and submit a list of responses to reviewer comments as a separate submission item.

Specifically, please address the question related to the relatively high N deposition value compared to the treatments and the treatment uncertainty from reviewer 1. I would also like to hear your responses regarding the time for observing the treatment impact for ecosystem studies from reviewer 2.

All changes in the revised manuscript must be highlighted in Word to assist checking of revisions by the editorial staff.

Reviewer 1 ·

Basic reporting

(1) In the whole article, almost all the unit of N addition rate are wrong. For example, in line 9, the g N-2 y-1 is wrong. Please check and revise throughoutly.
(2) Some literature references, agricultural ecosystems, are not closely related to the topic of this study. For instance, lines 57-63, 88-93---
(3) The figures are not appropriately shown, for example the different colours of the columns are lack of enough distinction. And the lowercase and uppercase letters should be exchanged in order to make a more understandable expression.
(4) Please give the elevation and slopethe of the study area in detail, because these two variables are important to explain the results.

Experimental design

(5) In lines 139-142, the authors said that the levels of N addition rate were designed based on the national mean N deposition rate (2.11) reported by Lü & Tian (2007). First, I think a more reliable value is needed according to references (also includs chinese papers). Second, the year 2007 is long-time ago to support your design. Third, the mean N deposition rate (2.11) is a relative high value which can potential affect the result under the current treatments (0,3,6,9). Please explain or discuss the uncertainty.
(6) Please indicate the distribution of the plots. Randomly? And how to treat the litter layer?

Validity of the findings

(7) In fig. 1, please explain why the micro-aggregates have the highest contents of T-GRSP?
(8) In fig. 2, please explain why the SOC increased so much under the three N addition treatments only over two years?

Additional comments

line 12, 'clay–silt aggregate'? is this correct?
line 15, 'variable' is not a clear word
lines 23-24, Redundant sentences. no clear message
line 32, 'negative effects' All negative below?
line 35, available N? what does it mean?
lines 57-63, delete because these are not closely related.
line 73, EE-GRSP, full name
line 74, T-GRSP, full name
line 128, 'surface soil' mineral soil? in detail of the depth.
line 130, hm2? should be ha?
line 149, why choose 0-20 cm? Does this include the litter layer?
Table 2, mean weight diameter (MWD)? or the aggregates distribution?

Reviewer 2 ·

Basic reporting

Overall the paper is well written and meets the standards of the paper, but it could benefit for another round of editing. There are a few places in the text were the sentences structure needs editing and also place were letters are missing in words. It is possible that I did not catch everything so please take some time to go through it again.
Line 7 - “ changes soil aggregate stability when it alters the” could be clarified by writing “ changes soil aggregate stability by altering the”
Line 30 – also the clarity here could benefit from some editing by changing “rapid increase in global nitrogen (N) deposition, which is expected to increase by 70% in 2050 than in 2000” to “rapid increase in global nitrogen (N) which is expected to increase by 70% in 2050 compared to 2000”
Line 31 – capital T is needed after punctuation
Line 33 – space needed between “rates” and parentheses
Line 39 - “these aggregate” - needs an “s” at the end
Line 105 – First sentence beginning with “However” is not needed.
Line 131 – the average height of the tree is probably not “11.2 mm”
Line 151 – add capital letter “T” after punctuation
Line 154 – instead of “filtered through” it might be better to use “sifted through” or “passed through”
Line 155 - plot instead of “pot”
Line 196 – add “in all samples” at the end of the sentence before the parenthesis.
Line 280 – remove “high because”
Line 311 – add “to” between “fails support”

Literature references:
The paper is very well referenced and the author does a nice job giving credit to a diverse group of authors. One of the references is a little old – would it be possible to find a reference of a newer date? Obviously some times that is not possible and many older references are key to the research conducted in this paper.
The reference eluded to is – Line 31 – Galloway et al., 2004
Dirnböck, T., Foldal, C., Djukic, I. et al. Climatic Change (2017) 144: 221. https://doi.org/10.1007/s10584-017-2024-y might not have the exact year estimates, but adds a nice focus on climate change and therefore supports the importance of the understanding of the effect of N deposition.

The introduction is generally well written and gives the reader the overall level of information that is needed to understand the importantance of the paper and the questions stated.
Improvements in the understanding of the methods used for aggregate stability would be of great benefit to the reader. In line 74 – the mean weight diameter is mentioned, but not thoroughly introduced. Please explain what MWD measures and why this method is used.

It is easy to understand the importance of the study from the introduction, but it might be helpful for the reader if the two hypotheses were stated in line 119 as a way to wrap up the introduction. That way they are clear from the beginning and it makes it easier to follow the rest of the paper.
The last sentence in the introduction is very bold. The data represented in this paper does not reveal the underlying mechanism for soil aggregation stability in response to N addition. Please rewrite this concluding sentence to more accurately summarize the overall goal with this study.

Experimental design

The research fits with in the scope of the journal and the question is well defined. It is a field were more research is needed and it would be nice to see this study continued beyond 2 years to see what effect time will have on the aggregate stability. For ecosystem studies such as this, 2 years is when you might expect to start seeing changes, but aggregate formation might be much slower process in a natural ecosystem than what has been seen in cultivated fields therefore 3-5 years should be considered.
The description of the study area is impressive and hopefully all the data collected on the biomass can also be use in a long term study on the effect of N deposition on plant community composition, growth rate of trees etc.
A few edits could improve the understanding of the experimental design
Line 144 – 147 – it is not clear if a sample was collected before addition of N. Please also clarify if samples were collected in 2016. A re-write of this section would make the time line for collection of samples more clear.
Line 149 – saying “In September 2015” makes it sound like that was the only time samples were taken. Please clarify.
Line 151 - no need to say “First” and the line 152 – starting with Second is not needed
Line 159 - it might be worth changing this sentence to saying - “samples (100 g) placed in a 250 um sieve and submerged in deionized water for 5 min”
Methods:
It is not clear to the reader why pits needed to be dug. Sampling from above using a soil corer would seem simpler than to create a larger disturbance in a fairly small plot area. What was used to “drill” a sample? What was the volume sampled? What is meant by “along natural fissures”. Please clarify the sampling method.
Also in line 155 – how were the samples stored?
Maybe add in line 171 – that it was the pellet that was re-extracted 5 times.
The issue with glomalin extraction is often that it can not be separated from total protein using the Bradford method. The two papers listed below address this issue and one also suggest using another method for detection. It would strengthen the discussion to talk about the bias in the method – and that the increase in T-GRSP could be from an increase in other proteins than glomalin. Take a look at these two papers and incorporate them in the discussion as suggested above. Roberts and Jones, 2008. Critical evaluation for determination of total soil protein in solution. Soil Biology and Biochemistry Vol 40 Issue 6. 1485-1495; Zbiral et al., 2017. Determination of glomalin in agriculture and forest soils by near- infrared spectroscophy. Plant Soil Environ. Vol 63, No. 5: 226-230.

Validity of the findings

The results section is well written and complete. The use of statistics in the analysis is appropriate and well done, but the conclusion in writing is some times contradicting the results described from the analysis. In the section from line 196-201 the author first describes that there is no significant difference between samples and then in the next line differences are described. Line 199-201 should not be included since there is no significant difference between the samples and therefore the pattern described is not valid. In line – 217 another set of data is described where not all of them are significantly different from each other and can therefore not be listed as such.
Three significant digits are provided for the results listed in line 204, 216, 239, 240 – are the methods used in this study sensitive enough to provide that kind for resolution?

In the discussion in line 281-285 speculations are made why the decrease in EE-GRSP drops while the T-GRSP continues to accumulate. From this study it can not be concluded that the T-GRSP will not continue to go up and the possibility that it will stay high or increase should also be discussed. As mentioned above the measurement of total protein in the soil might not be able to be distinguished from the T-GRSP. Please consider the llimitations of the method used and the possibility of an increase in other soil proteins.
In line 296-297 – could it also be that decomposition rates are increase not just plant growth?

Additional comments

It was a pleasure to read the paper and I look forward to seeing more papers linked to this study to answer some of the questions posed.
Continuing the study for 3-5 more years would be highly recommended and to experiment with using near-infrared spectroscopy for analysis of the glomalin would be interesting. Other methods such as microbial ecosystem analysis and respiration rate would add to the understanding of the complexity of the effect of N deposition. And so would analysis of plant community composition and growth rate of trees over time. Hopefully similar studies will be conducted other places in the world so that we can gain a better understanding how the soil type, texture class, OM content etc. of soils play a role in the effect of N deposition and aggregate stability. Good luck with future investigations!

---

## Round 0.2 · accepted · Accept

I have received your revision. After discussing with the external reviewer, I have decided to accept your manuscript for publication in PeerJ.

# Reviewer 1 ·

Basic reporting

The revision is acceptable.

Experimental design

The revision is acceptable.

Validity of the findings

The revision is acceptable.